# Severely Damaged Freeze-Injured Skeletal Muscle Reveals Functional Impairment, Inadequate Repair, and Opportunity for Human Stem Cell Application

**DOI:** 10.3390/biomedicines12010030

**Published:** 2023-12-21

**Authors:** Daniela Fioretti, Mario Ledda, Sandra Iurescia, Raffaella Carletti, Cira Di Gioia, Maria Grazia Lolli, Rodolfo Marchese, Antonella Lisi, Monica Rinaldi

**Affiliations:** 1Department Biomedical Sciences, Institute of Translational Pharmacology, National Research Council, Area di Ricerca Roma2 Tor Vergata, 00133 Rome, Italy; mario.ledda@ift.cnr.it (M.L.); sandra.iurescia@ift.cnr.it (S.I.); mariagrazia.lolli@yahoo.it (M.G.L.); antonella.lisi@ift.cnr.it (A.L.); 2Department of Translational and Precision Medicine, Sapienza University of Rome, 00185 Rome, Italy; raffaella.carletti@uniroma1.it; 3Department of Radiological, Oncological and Pathological Sciences, Sapienza University of Rome, 00161 Rome, Italy; cira.digioia@uniroma1.it; 4Department of Clinical Pathology, FBF S. Peter Hospital, 00189 Rome, Italy; marchese.rodolfo@fbfrm.it

**Keywords:** traumatic muscle injury, mouse model, freeze injury, physical impairment inflammatory response, muscle regeneration, cytosolic DNA sensing, satellite cells, human amniotic mesenchymal stem cells

## Abstract

Background: The regeneration of severe traumatic muscle injuries is an unsolved medical need that is relevant for civilian and military medicine. In this work, we produced a critically sized nonhealing muscle defect in a mouse model to investigate muscle degeneration/healing phases. Materials and methods: We caused a freeze injury (FI) in the biceps femoris of C57BL/6N mice. From day 1 to day 25 post-injury, we conducted histological/morphometric examinations, an analysis of the expression of genes involved in inflammation/regeneration, and an in vivo functional evaluation. Results: We found that FI activates cytosolic DNA sensing and inflammatory responses. Persistent macrophage infiltration, the prolonged expression of eMHC, the presence of centrally nucleated myofibers, and the presence of PAX7+ satellite cells at late time points and with chronic physical impairment indicated inadequate repair. By looking at stem-cell-based therapeutic protocols of muscle repair, we investigated the crosstalk between M1-biased macrophages and human amniotic mesenchymal stem cells (hAMSCs) in vitro. We demonstrated their reciprocal paracrine effects where hAMSCs induced a shift of M1 macrophages into an anti-inflammatory phenotype, and M1 macrophages promoted an increase in the expression of hAMSC immunomodulatory factors. Conclusions: Our findings support the rationale for the future use of our injury model to exploit the full potential of in vivo hAMSC transplantation following severe traumatic injuries.

## 1. Introduction

Skeletal muscle architecture is a primary determinant of its mechanical behavior. Traumatic injuries to the musculoskeletal system involving critical loss of muscle mass are well beyond the muscle’s inherent capacity for self-repair, leading to long-term deficits in muscle structure and strength [1,2,3]. The consequences for the affected patients are often detrimental, causing severe physical impairment that leaves the patient in extreme pain and discomfort, often with a life-long disability [4,5]. The regeneration of severe muscle injuries is an unsolved medical need and is a topic of considerable scientific interest in both civilian and military medicine.

Muscle regeneration after an acute injury follows a strict schedule consisting of the pro-inflammatory phase, the resolution of inflammation, and the restorative phase. The inflammatory response that occurs right after damage is central in the coordination of muscle injury response and regeneration. The damaged myofibers undergo necrosis upon muscle tissue injury, releasing intracellular contents and chemotactic factors into the lesion site. Intracellular contents act as damage-associated molecular patterns (DAMPs) that activate and recruit immune cells to the site of trauma [6,7]. Rapid tissue-resident macrophage activation and leukocyte recruitment occur in the damaged muscle environment, enriched with pro-inflammatory cytokines such as interferon (IFN)-γ and tumor necrosis factor (TNF)-α [8]. Shortly after neutrophil recruitment, circulating monocytes and macrophages are recruited to the damaged tissue, where phagocytic macrophages, skewed towards an “M1-like” pro-inflammatory phenotype, are responsible for the clearance of necrotic cells and debris, including DNA [9].

Inflammation is coupled to the initial phases of myogenesis during the early stages of muscle regeneration. The inflammatory microenvironment and infiltrating immune cells trigger signaling cascades that regulate the activation and proliferation of a pool of PAX7^+^ muscle-resident stem cells/satellite cells (MuSCs). The activated MuSCs can benefit from the support provided by mesenchymal stem cells (MSCs) derived from bone marrow [10,11]. In response to injury signals, these cells, acting as inflammatory sensors, move from their niche into the peripheral circulation and reach target tissues, where they participate in the regenerative process via the paracrine secretion of regulatory molecules that interact with innate immune system components [12,13,14,15]. The immunomodulating capacity of the MSCs, in turn, depends on a proper “licensing” by inflammatory milieu [16,17,18]. Therefore, the dynamic regulatory crosstalk between MSCs and macrophages is critical in triggering the switch from an inflammatory to an anti-inflammatory status and promoting the generation of a more favorable microenvironment for tissue repair [19,20]. For this reason, manipulating the inflammatory response to muscle injury can be used to improve regeneration.

A wide variety of pre-clinical models of traumatic muscle injury have been developed [21,22]. The model chosen in this investigation was the freeze-based skeletal muscle injury, which mimics severe traumatic muscle injuries that can destroy nearly all fibers at the site of injury [23,24]. The resulting nonhealing muscle defect leaves deficits in muscle structure and strength, causing chronic functional impairment. In this work, we induced a freeze injury (FI) and evaluated its outcome in a mouse model to investigate the phases of muscle degeneration/healing at morphologic, molecular, and functional levels; the study’s results, in particular, add to the body of knowledge regarding inflammatory and anti-inflammatory responses to a traumatic injury.

We intentionally investigated the inflammatory response on a transcriptional level, even considering the innate immune response after the FI. Our injury model can be helpful in evaluating the potential of stem-cell-based therapy in improving the outcome of inflammatory response and repair following severe traumatic injuries. Thus, the modulation of the expression of genes involved in inflammatory response was investigated in vitro by means of assessing the crosstalk between murine macrophages and human mesenchymal stromal/stem cells from amniotic tissue (hAMSCs). These cells are considered an excellent candidate in cell-therapy-based treatments due to their low immunogenicity, multilineage differentiation potential, plasticity, and immunomodulatory properties [25]. For the first time, we analyzed the immunomodulatory potential of M1-polarized murine macrophages in the expression of hAMSC mRNA factors correlated with immunomodulation/inflammation processes. Overall, our findings will help provide insight into the inflammatory environment associated with this model of severe muscle injury, supporting the rationale for future in vivo hAMSC transplantation.

## 2. Materials and Methods

### 2.1. Animals

Forty-eight healthy male C57BL/6N mice (Charles River Italia) weighing between 22 and 24 g with an average age of ten weeks were used in this study. The animals were maintained at 22  ±  2 °C under a 12 h–12 h light/dark cycle with 50–60% humidity for at least one week before the experiment. Mice’s weight was measured at each time point to verify animal health conditions (Appendix A).

### 2.2. Freeze Injury

After acclimation, 44 animals were fully anesthetized by intraperitoneal injection of Zoletil 100 (50 mg/mL tiletamine HCl + 50 mg/mL zolazepam HCl; Virbac, Milan, Italy) and Rompun 20 (20 mg/mL xylazine; Bayer S.p.A., Milan, Italy). The right hind limb of each mouse was shaved, and muscle was exposed via a 1 cm long incision in the aseptically prepared skin overlying the *biceps femoris* muscle. A metallic probe of 4 mm diameter was cooled in liquid nitrogen and then applied vertically to the exposed muscle for 10 s with a moderate uniform pressure to cause severe damage [24,26]. The skin was sutured with a 6-0 suture string (Assut Europe, Rome, Italy), and mice were kept warm until awake. Buprenorphine (0.1 mg/kg; diluted to 0.015 mg/mL; Sigma-Aldrich, Merck KGaA, Germany) was administered subcutaneously before recovery from anesthesia. In the first few days after the injury, wet food placed onto cage substrate was provided as a supplement in addition to pelleted food and water. This arrangement has made it easier to control the gait and the physiological state of animals. Mice’s weight was measured at each time point to verify animal health conditions. The control group (4 mice) and each experimental group (4 mice) at 1, 2, 4, 7, 12, and 25 days were euthanized using CO_2_ asphyxiation. The control group comprises untreated animals.

### 2.3. In Vivo Functional Evaluation

Motor behavior assessments were performed on four randomly selected animals per group. Each mouse was placed on the rotarod (Ugo Basile s.r.l, Varese, Italy), which accelerated from 5 to 50 rpm (0.5 cm/s every 5 s) over the trial time of 120 s. Trials terminated when animals fell off or the maximum time was reached. A trip switch on the floor below was set to record the latency until the mouse fell from the rotating rod. Each mouse was tested three times on separate trials. The control group was considered the reference control with 100% functionality. The balance beam test assessed the motor coordination and balance of injured mice. The beam apparatus consists of a 1 m beam with a flat surface of 15 mm resting 40 cm above the table top on two poles. A dark box was placed at the end of the beam. Food and nesting material from home cages were placed in the black box to attract the mouse to the finish point. A lamp was used to shine light above the start point and served as an aversive stimulus. During a single day of testing, each mouse was given three trials with a minimum of 10 min between the trials. A test trial was given to familiarize the mouse with the beam. For each trial, the mouse was placed at the beginning of the beam, on the opposite end from where the dark box was placed. The measurements recorded the number of paw faults, defined as the number of times the hind legs slide off the horizontal surface of the beam.

### 2.4. Tissue Sampling

Immediately after euthanasia, the injured thigh of the hindlimb was dissected at 1, 2, 4, 7, 12, and 25 days post-injury. The healthy thigh tissue of four uninjured mice was used as tissue control. Dissected hindlimbs of 20 mice were immediately frozen in liquid-nitrogen-chilled isopentane and stored at −80 °C until molecular and tissue analyses on damaged areas were performed. In addition, hindlimbs from 24 mice were placed in a 10% neutral-buffered formalin solution, embedded in paraffin, and used for light microscopic examination and morphometric analysis. Four animals were randomly assigned at each experimental time point.

### 2.5. Qualitative Histological Observation

Five-micrometer cryostat sections of injured thighs dissected and frozen at 4 and 7 days post-injury were stained with hematoxylin and eosin (H&E). Ten images at 10× magnification were selected to evaluate the changes in damaged muscle tissue.

### 2.6. Histological and Morphometric Analysis of Injured Muscle Tissue

Formalin-fixed and paraffin-embedded muscle tissue sections (3 μm), obtained from control and injured mice, were stained with H&E and Sirius Red, a collagen-specific stain. Changes in muscle tissue related to damaged and perilesional area at each experimental time point were assessed by light microscopic analysis. For each muscle tissue, sections consisting of the largest damaged area and perilesional one, i.e., the area immediately next to the damaged area (maximum distance of 200 µm), were selected for histological and morphometric analysis. The extension of the damaged area (mm^2^) was evaluated using the software Aperio Image Scope (version 12.3.3; Leica Biosystems, Buccinasco, Milan, Italy) as the ratio between the damaged area and the total area of the muscle tissue section and expressed as a percentage. A quantitative evaluation of muscle cells with internal nuclei, often centrally positioned, in the regenerating area from day 4 to day 25 post-injury was performed in 5 fields at 20× (magnification) with the transverse section of the muscle fibers and expressed as the mean. The morphometric evaluation of muscle tissue changes was performed on digital images captured after scanning H&E and Sirius Red stained slides with an Aperio scanner (version 12.3.3: Leica Biosystems, Buccinasco, Milan, Italy). Ten selected images at 20× of H&E-stained slides were analyzed to measure the cross-sectional area of muscle cells (CSA, μm^2^, used to evaluate muscle fiber sizes based on fiber area in cross-section) using computerized imaging software (ImageJ-win32, NIH, Bethesda, MD, USA) in 100 muscle cells transversally sectioned and this was expressed as the mean [27]. The consecutive muscle tissue sections (3 μm) stained with Sirius Red were used for morphometric analysis of red-stained interstitial collagen volume fraction with computerized imaging software (Image J-win32, NIH, Bethesda, MD, USA) and expressed as a percentage [28].

### 2.7. Evaluation of Inflammatory Infiltrate Cells

The inflammatory infiltrate cells, mainly neutrophils and mononuclear cells, were evaluated in formalin-fixed and paraffin-embedded tissue sections at each experimental time point. In muscle tissue sections stained with H&E, the number of neutrophils was manually counted and expressed as the ratio between number of neutrophils and total damaged area, defined as mean value/total area. The immunophenotype characterization of inflammatory infiltrate cells was evaluated by immunohistochemistry. Consecutive formalin-fixed and paraffin-embedded tissue sections (3 μm) were deparaffinized in xylene and rehydrated through a graded alcohol series. Endogenous peroxidase activity was blocked by 3% hydrogen peroxide. The sections were treated by boiling in citrate buffer (0.01 mol/L, pH 6) in a microwave (750 W) and incubated overnight at 4 °C with primary antibody: anti-CD68 (1:500, anti-mouse rat monoclonal antibody (FA-11), ab5344, Abcam, Cambridge, UK), anti-CD163 (1:500, anti-mouse rabbit monoclonal antibody (EPR19518), ab 182422, Abcam, Cambridge, UK), and anti-CD206 (mannose receptor, 1:500, anti-mouse rabbit polyclonal antibody, ab64693, Abcam, Cambridge, UK), respectively. The reaction product was amplified by an Ultra Tek HRP Staining System (Scy TeK Laboratories, Logan, UT, USA) and visualized with 3,3′-diaminobenzidine (Dako, Glostrup, Denmark). The slides were counterstained with Mayer’s hematoxylin. Negative controls were obtained by omitting the primary antibody. All immunostained sections were analyzed using a Leica optical microscope and then captured with an Aperio scanner (version 12.3.3; Leica Biosystems, Buccinasco, Milan, Italy). The digital images of the whole damaged area were analyzed to evaluate the number of positive cells with computerized imaging software (ImageJ-win32, NIH, Bethesda, MD, USA) and this was expressed as the ratio of the number of positive cells to the total damaged area, defined as the mean value of positive cells/total area.

### 2.8. IHC Identification of Satellite Cells PAX7^+^ Satellite Cells

Consecutive formalin-fixed and paraffin-embedded muscle tissue sections (3 μm) were used to evaluate the presence of satellite mononucleated cells in the adjacent periphery of the lesioned area. The endogenous peroxidase activity was blocked by 3% hydrogen peroxide. The sections were treated by boiling in citrate buffer (0.01 mol/L, pH 6) in a microwave (750 W) and incubated for an hour at room temperature with anti-PAX7 antibody (1:100, anti-mouse monoclonal, sc81975, Santa Cruz Biotechnology, Inc., Dallas, TX, USA). A Mouse-on-Mouse Polymer IHC Kit (ab127055, Abcam Cambridge, UK) was used to label the primary antibody. The reaction product was visualized with 3,3-diaminobenzidine (Dako, Glostrup, Denmark). The sections were counterstained with Mayer’s hematoxylin. Negative control was obtained by omitting the primary antibody. All immunostaining slides were captured with an Aperio scanner (version 12.3.3: Leica Biosystems, Buccinasco, Milan, Italy). Ten images at 40× magnification were selected to evaluate the number of positive cells with computerized imaging software (ImageJ-win32, NIH, Bethesda, MD, USA) and this was expressed as a percentage (ratio of the number of positive nuclei to muscle cell nuclei) [29].

### 2.9. RNA Isolation and Quantitative RT-PCR

Cells or unfractionated muscle tissues were collected and transferred into EXTRAzol (Blirt S.A., Gdańsk, Poland) in 2 mL Eppendorf tubes for homogenization. After chloroform phase separation, supernatants were moved into the Direc-zol^TM^ RNA MiniPrep Plus columns (Zymo Research, Irvine, CA, USA), and total RNAs were extracted following the manufacturer’s instructions. Genomic DNA contamination was removed by DNase I treatment directly on the minikit column. Purified RNA was quantified using Eppendorf BioSpectrometer^®^ Kinetic (Eppendorf AG, Hamburg, Germany); 1 μg of RNA was reverse-transcribed using an iScript^TM^ cDNA synthesis kit (Bio-Rad, Hercules, CA, USA) according to the manufacturer’s instructions. Quantification of all gene transcripts was carried out by reverse transcription (RT) quantitative polymerase chain reaction (qPCR), using SsoAdvanced^™^ Universal SYBR^®^ Green Supermix (Bio-Rad, Hercules, CA, USA) and a CFX Connect^TM^ Real-Time PCR System (Bio-Rad, Hercules, CA, USA). SYBERGreen gene expression assays (BioRad, Hercules, CA, USA) were used to quantify target genes controlling IRF3-dependent cytosolic DNA sensing (*Ifnβ*, qMmuCED0050444; *Cxcl10*, qMmuCED0001068; *Ifit*, qMmuCID0012621; *Ifit2*, qMmuCID0026327; *Ifit3*, qMmuCID0041372; *Oasl2*, qMmuCED0047823; *Irf7*, qMmuCED0040274; *Rsda2*, qMmuCED0044620; *Tmem173*, qMmuCED0046127; *Gapdh*, qMmuCED0027497). Primers used for detecting genes involved in muscle regeneration and the inflammation process are reported in Appendix A. Gene expression quantification was performed using the ΔCt method, and fold change values were reported relative to *Gapdh* mRNA.

### 2.10. RAW 264.7 Macrophage Culture

The RAW 264.7 murine macrophage cell line was obtained from the American Type Culture Collection (ATCC, TIB-71™, Rockville, MD, USA). Cells were grown in high-glucose Dulbecco’s modified Eagle’s medium (DMEM; Euroclone, Milan, Italy), supplemented with 10% heat-inactivated fetal bovine serum (FBS, Euroclone, Milan, Italy), 2 mM glutamine (Sigma, Darmstadt, Germany), 100 U/mL penicillin (Sigma, Darmstadt, Germany), and 100 μg/mL streptomycin (PBI international, Milan, Italy). Cells were cultured on a plastic Petri dish at 37 °C in a humidified incubator containing 5% CO_2_. Cells were passaged after reaching 90% confluence, detached with trypsin-EDTA-0.25% (Sigma, Darmstadt, Germany), and subcultivated at a 1:6 ratio in T-75 flasks.

### 2.11. Isolation and Culture of hAMSCs

hAMSCs were isolated from term human placenta of healthy women after written informed consent, following the protocol reported in Ledda et al. [30]. Briefly, hAMSCs were obtained by a two-step procedure: fragments of the amniotic membrane were incubated for 1 h with 0.25% trypsin-EDTA solution to remove human amniotic epithelial cells (hAECs), then the supernatant was discarded and the remaining mesenchymal cells underwent a second digestion with 0.1% collagenase IV (Sigma-Aldrich, Saint Louis, MO, USA) and 20 µg/mL DNAse I (Sigma-Aldrich, Saint Louis, MO, USA) solution in DMEM for 3 h. The supernatant was neutralized with FBS and then centrifuged at 1200 rpm for 10 min. Each pellet was suspended in complete culture medium containing: DMEM, 10% FBS, penicillin (100 U/mL) and streptomycin (100 μg/mL) (PBI international, Milan, Italy), EGF (10 ng/mL, ImmunoTools, Friesoythe, Germany), and β-mercaptoethanol (55 µM, Sigma-Aldrich, Saint Louis, MO, USA). HAMSCs were cultured on plastic Petri dishes and maintained at 37 °C in a humid atmosphere containing 5% CO_2_.

### 2.12. Transwell Co-Culturing

First, 15 × 10^3^ hAMSCs were seeded on 6-multiwell plastic plates and left to adhere overnight; at the same time, 70 × 10^4^ RAW 264.7 macrophages were plated on the upper compartment of permeable polyethylene terephthalate (PET) Transwell inserts with pore size 0.4 μm (Corning, Corning, NY, USA) and stimulated with 100 ng/mL of LPS. After 24 h, the medium was replaced for both cell types, and they were added to a co-culture of hAMSCs (seeded on the lower compartment of multiwell) and stimulated RAW macrophages (seeded on the upper compartment of transwell inserts). After 24 h of co-culture, hAMSCs or RAW264.7 were detached and RNA was extracted.

### 2.13. Statistical Analysis

Statistical analyses were performed on MedCalc software (version 20.015), using Student’s *t*-test and ANOVA with Tukey’s post hoc test. Significance was accepted at *p*  ≤  0.05 and *p*  ≤  0.001. Quantitative data are displayed as mean  ±  standard deviation.

## 3. Results

### 3.1. Freeze Injury Induction and Outcome on Motor Performance

Considering the protocols already used to obtain damaged muscle in vivo models by freezing [24,26,31,32], we created a traumatic localized muscle injury in vivo model by performing a cryoinjury on the hind limb of C57BL/6N mice. After skin incision and muscle exposition, a stainless-steel metal rod with a flat surface of 4 mm diameter was cooled in liquid nitrogen for 10 min and then applied vertically to the exposed *biceps femoris* (BF) muscle for 10 s with a moderate uniform pressure. Qualitative histological observation of the entire freeze-injured muscle showed severe damage characterized by substantial inflammatory infiltration. Representative images of hematoxylin- and eosin-stained cross-sections of mouse BF muscles are shown in Appendix A.

Loss of muscle function after injury was assessed in vivo by motor coordination and balance tasks to detect injury-induced deficits. The average latency to fall is represented in Figure 1A and served as an indicator of motor coordination.

Each mouse was tested three times on separate trials using an accelerating version of the test, where speed increases were incorporated. The results revealed that our FI protocol produced significant deficits in the animal’s ability to perform the rotarod task; the deficit due to the damage persisted until the end of the experiment (day 25) (Figure 1A). The mean time taken by injured mice was less than 47 s in contrast to the control (uninjured) group score, with an average time of 63/64 s. The balance beam test assessed the motor coordination and balance of injured mice. Performance on the beam was quantified by measuring the number of paw slips that occurred in the process. Nesting material from home cages and food were placed in the black box to attract the mouse to the finish point. A lamp was used to shine light above the start point and served as an aversive stimulus. The values relating to paw faults from day 1 to day 25 in all injured mice showed an evident inability to walk without slipping (Figure 1B), demonstrating a durable disability (day 7, *p* = 0.00067 and day 25, *p* = 0.00165). Despite the moderate functional improvement observed on day 4, probably due to the reduction of the tissue edema, these results demonstrated an incomplete functional recovery.

### 3.2. Histological and Morphometric Analysis of Freeze-Injured Muscle Tissue

The histological analysis documented muscle damage’s presence and evolution in hindlimb tissue samples of all freeze-injured mice from day 1 to day 25 post-injury. Uninjured control samples were also examined and showed healthy skeletal muscle myofibers (Figure 2A).

The extension of the injured area at day 1, day 2, day 4, and day 7 post-injury, evaluated by Image Scope, was 17.40% ± 14.80, 35.66% ± 13.27, 28.42% ± 22.32, and 31.83% ± 17.08, respectively (Figure 2B). Injured samples were characterized by necrosis of myofibers and interstitial edema with infiltrating inflammatory cells (Figure 2A “injured area”). At day 12 and day 25 after injury, the damaged area was 18.66% ± 5.50 and 19.25% ± 5.67, respectively (Figure 2B), and was characterized by connective tissue replacement (i.e., replacement fibrosis) (Figure 2A) and still showed the presence of macrophages. Fibrosis quantification was determined by staining tissue slides with Sirius Red. The control tissue showed the expected interstitial collagen amount (Figure 2A, Sirius Red, uninjured). Figure 2C represents the percentage of the injury area occupied by connective tissue. Collagen deposition significantly increased in all injured animals compared with uninjured tissue (*p* < 0.05) and, most importantly, from day 7 to 25 post-FI compared with day 4 (*p* < 0.05; Figure 2C). The perilesional area appeared largely unaffected (Figure 2A, Sirius Red). This increase in collagen content indicated migration of fibroblasts into the site of injury and replacement fibrosis, which is still persistent 25 days post-injury (Figure 2A, Sirius Red and Figure 2C).

To evaluate the degree of muscle regeneration, we assessed the size of the myofibers, their nuclei location, and the percentage of Pax7^+^ satellite cells in the injured samples. The measured myofiber cross-sectional area (CSA) was first used to obtain the histograms in Figure 3A,D.

At each time point after FI, myofibers in the injured samples showed CSA values significantly smaller than those of myofibers present in both areas surrounding the injury and control tissue. The size distribution of the myofibers further indicated that the number of fibers with small CSA in the regenerating area is greater than in the perilesional area (Figure 3B,E). At 4 days post-FI, most myofibers showed a CSA value below 500 μm^2^, indicating de novo myofiber formation. Although regeneration had been occurring, the frequency distribution plot at 25 days post-FI also revealed that the CSA of regenerating myofibers did not reach the same value as myofibers in uninjured mice (Figure 3B). In the area surrounding the injury, the CSA of myofibers does not change over time (Figure 3E).

Newly formed myofibers with centrally located nuclei were also quantified. Our results showed that smaller myofibers with internal nonperipheral nuclei were present starting from day 4 after FI onwards (Figure 3C). Twelve days after the injury, there were significantly more muscle fibers with centrally located nuclei than in the uninjured control. A significant number of centrally nucleated myofibers remained 25 days after injury (Figure 3C).

Antibody immunostaining showed several PAX7^+^ cells localized in the area surrounding the lesion. After the injury, a statistically significant number of PAX7^+^ cells were observed at each time point compared with uninjured muscle. The percentage of PAX7^+^ cells was particularly high on day 4 post-injury (*p* < 0.001; Figure 3F). These results suggested satellite cell activation and proliferation during the muscle regenerative process.

### 3.3. Freeze Injury Activates IRF3-Dependent Cytosolic DNA Sensing and Inflammatory Response

Necrotic-cell-derived DNA can be recognized as a DAMP by pro-inflammatory phagocytic macrophages and activates the cytosolic DNA sensing pathways involving the cyclic GMP–AMP synthase (cGAS) and the stimulator of interferon genes (STING) protein [33]. RT-qPCR was performed to evaluate transcript abundances of the pathway effector STING and the transcriptional targets of the cGAS-STING signaling pathway (Figure 4).

We first assessed the induction of *Ifnβ1* and *Cxcl10*, which represent IRF3-driven pro-inflammatory cytokines and chemokines [34,35]. Significantly increased *Ifnβ1* mRNA levels were detected as early as day 1 post-injury, as expected for a prototypical IRF3-driven gene; thereafter, its expression rapidly decreased, reaching an expression level similar to that of uninjured muscles. A significant upregulation of *Cxcl10* expression was also observed on both day 1 and day 4 after damage compared with uninjured controls. Among IRF3-responsive genes, transcript levels of IFN-induced protein with tetratricopeptide repeat 1 (*Ifit*1), *Ifit*2, and *Ifit*3 and 2′-5′-oligoadenylate synthase-like 2 (*Oasl2)* [35,36] were significantly increased in comparison to uninjured muscle. We observed higher *Ifit*1, *Ifit*2, and *Ifit*3 transcription levels in injured muscles than controls, with *Ifit*2 and *Ifit*3 reaching peak expression at day 4 after FI. Comparable levels of *Oasl2* transcripts were observed on day 1 and day 4 after FI, while its expression significantly decreased on day 7 after injury. Upregulated genes such as *Irf7* are prototypical interferon-stimulated genes (ISGs) whose transcription is activated by *Ifnβ*1. Accordingly, a marked increase in *Irf7* expression level is achieved 1 day after muscle damage; the mRNA levels remained relatively constant on day 4, whereas they were significantly decreased on day 7. *Rsda*2 transcript levels were significantly increased at both 1 day and 4 days after injury, reaching an expression peak at 4 days. We also observed a significant increase in *Tmem*173 mRNA coding for STING, the key mediator in the immune response to cytoplasmic DNA. The induction of the *Tmem*173 gene expression reached the maximum level at day 1 post-injury, then the transcript levels declined over time. At day 7, after damage, transcription levels of most of the genes appeared to decline without ever reaching control levels. Together, these results indicated that DAMP sensing by phagocytes present in the injured tissue induced IRF3-dependent type I IFN transcriptional response, thus fueling inflammatory response after local FI.

### 3.4. Inflammatory Infiltrate in FI-Damaged Muscle Tissue

The influx of inflammatory cells is clearly detectable and restricted to the FI-damaged region (Appendix A). Early invasion of injured muscle by immune cell populations was mainly due to neutrophils. They were the most abundant inflammatory cells in the damaged muscle at an earlier time point (day 1) and showed a significant progressive decrease from day 2 to day 7 after injury (*p* < 0.05). No neutrophils were observed on day 12 after FI and until the end of the observation period (Figure 5A).

Injured muscle sections also showed macrophage recruitment detected by CD68 staining, whereas macrophages were virtually absent in uninjured tissue (Figure 5B,C). The immunohistochemical (IHC) analysis of infiltrating macrophages showed positive staining for expression of the scavenger receptors CD163 and CD206, indicating their alternative M2-biased activation (Figure 5D). At day 25 post-injury, the CD206^+^ pro-resolving macrophages became the predominant population compared with CD163^+^ macrophages, highlighting that the CD206^+^ M2-biased macrophages play an essential role during the late post-injury stages as they support collagen production and deposition [37]. Although we observed an overall progressive decrease in CD68^+^ macrophage populations, injured muscle tissues showed macrophages present up to day 25 post-FI, indicating persistent infiltration (Figure 5B,C).

### 3.5. Pro-Inflammatory and Anti-Inflammatory Marker Expression in FI-Damaged Mice

M1 macrophages are involved in the early stage of inflammation and secrete high levels of pro-inflammatory cytokines such as tumor necrosis factor alpha (TNF-α), interleukin-6 (IL-6), and C-C motif chemokine ligand 2 (CCL2), also called monocyte chemoattractant protein-1 (MCP1). In contrast, M2 phenotypes are associated with the resolution of inflammation and tissue repair [38] and produce high levels of anti-inflammatory factors such as interleukin-10 (IL-10), macrophage mannose receptor 1 (CD206), and arginase 1 (Arg1). qRT-PCR analysis was used to evaluate markers of inflammation in injured muscle tissue. Pro-inflammatory Tnf-α, Il-6, and Ccl2 mRNA markers showed an early and statistically significant increase in their expression at day 1 post-injury, continuing to be upregulated until 25 days after damage compared with uninjured muscles (Figure 6).

The inflammatory response investigated at systemic levels showed an increased amount of IL-6 protein in the sera of the injured mice until the fourth day after the injury, indicating a possible systemic acute response (Appendix A). A significant increase in mRNA expression of Il-10, CD206, and Arg1 anti-inflammatory markers was also observed 1 day after injury, reaching an expression peak at day 4 and maintaining transcription levels higher than in uninjured control tissues at 7, 12, and 25 days (Figure 6).

### 3.6. Muscle Regeneration Marker Analysis in FI-Damaged Mice

The expression levels of the embryonic myosin heavy chain (eMHC), a marker transiently expressed in early regenerating myofibers, the transcription factor paired box 7 (Pax7), a marker for satellite cells, essential for regulating their expansion and differentiation during myogenesis, and Myod and Myog, transcription factors whose expression regulates myoblast differentiation by activation of muscle-specific genes, were studied through qRT-PCR analysis in FI-damaged mice. All regeneration marker genes exhibited a significant gradual increase in expression rate at day 1, day 4, and day 7 (*p* < 0.05) compared with uninjured control mice (Figure 7).

In particular, Myod, an early myogenic factor, showed a rising trend between days 1 and 7 and then decreased, reaching the initial mRNA control level. The regenerative eMHC and Myog markers showed a substantial increase on day 4 and maintained a higher expression than the control group until day 25. Statistically significant Pax7 mRNA upregulation compared with controls was observed at each time point, consistent with the immunohistochemical results described above. In injured mice, the modulation of myogenic markers suggests muscle regeneration process activation, confirmed by the early upregulation of inflammatory markers and the following anti-inflammatory response activation. Moreover, maintaining their high-level expression until the last time point (Figure 5 and Figure 6) provides direct evidence that muscle repair is ongoing.

### 3.7. Crosstalk between hAMSCs and Macrophages: Inflammatory and Anti-Inflammatory Marker Expression in M1 Macrophages

MSCs act as inflammatory sensors in acutely damaged muscle, interacting with macrophages, modulating their switch from a pro-inflammatory M1 into anti-inflammatory and pro-regenerative M2 phenotype and promoting regeneration by producing immunoregulatory, pro-angiogenic, and pro-regenerative factors [20]. To explore the therapeutic potential of muscle-regenerative protocols employing exogenous MSCs, we investigated the crosstalk between M1-polarized mouse macrophages and human mesenchymal stromal/stem cells from amniotic tissue (hAMSCs). To this end, we developed an in vitro model by co-culturing the RAW 264.7 mouse macrophage cells with hAMSCs using a Transwell assay [39]. First, we stimulated mouse M0 RAW 264.7 cells with lipopolysaccharide (LPS) to induce their classical transition toward the M1 phenotype [40,41]. The TNF-α and IFN-γ protein levels, secreted by RAW 264.7 cells stimulated for 24 h, were quantified by ELISA analysis to demonstrate the LPS effect. A statistically significant increase in both factors was observed in LPS-stimulated RAW 264.7 cells (M1 Mφ) compared with M0 RAW 264.7 cells alone (Appendix A). Afterward, we investigated whether M0 and M1 RAW 264.7 macrophages could be affected by co-culturing with hAMSCs. mRNA expression levels of the inflammatory markers, such as nitric oxide synthase 2 (Nos2), Tnf-α, and the anti-inflammatory marker Il-10, were analyzed at 24 h in co-cultured macrophages compared with control ones (M0 and M1 RAW 264.7 cells alone). qPCR analysis showed a statistically significant decrease in Nos2 and Tnf-α markers and a statistically significant increase in the Il-10 mRNA expression in co-cultured M1 macrophages. The expression levels of these markers did not present any changes in naïve M0 macrophages co-cultured with hAMSCs (Figure 8A).

### 3.8. Crosstalk between hAMSCs and Macrophages: Immunomodulatory Marker Expression in hAMSCs

To investigate the influence of M1 RAW 264.7 macrophages on hAMSCs, we studied the mRNA expression of markers correlated to immunomodulation/inflammation processes such as cyclooxygenase 2 (COX-2), a key enzyme involved in the synthesis of immunoregulatory factor PGE-2 strongly related to MSC immunomodulatory ability; indoleamine 2,3-dioxygenase 1 (IDO), an anti-inflammatory factor which plays a vital role in T-reg stimulation and in suppressing T-cell proliferation; and hepatocyte growth factor (HGF), an immune-modulatory factor which reduces the generation of inflammatory Th1 cells. The qPCR analysis showed that the COX-2, IDO, and HGF transcript levels in M1-primed hAMSCs were higher than those in the unprimed counterparts cultured in the presence of resting naïve macrophages, demonstrating that the immunomodulatory factors secreted by stimulated M1 macrophages can trigger hAMSCs to produce immunomodulatory molecules that are involved in the paracrine response (Figure 8B). On the contrary, COX-2, IDO, and HGF mRNA levels in co-cultures were unaffected by priming hAMSCs with unstimulated M0 macrophages (hAMSCs-naïve Mφ vs. hAMSCs). Together, these data showed the hAMSCs’ ability to downregulate the expression of Nos2 and Tnf-α inflammatory cytokines and enhance anti-inflammatory Il-10 marker expression in macrophages. hAMSCs and mouse M1-stimulated macrophages influence each other, increasing stem cells’ immunomodulatory potential by upregulating COX-2, IDO, and HGF mRNA levels.

## 4. Discussion

Herein, we have administered a freeze injury to a mouse model using liquid nitrogen that determined a critically sized nonhealing defect in the skeletal *biceps femoris* muscle. Cryoinjury is a well-described procedure generating delimited damage and necrosis with a discrete border between uninjured and injured muscle and reproducible outcomes, unlike other approaches using myotoxins and chemical agents [22,24,26,31]. Hardy et al. reported that FI represents an experimental model where most of the stem cells and their surrounding environment (basal lamina and vasculature) are destroyed, leaving in the zone of injury a “dead zone” devoid of viable cells [22]. The regenerative response can thus be reliably evaluated histologically by new myofiber formation as indicated by the cross-sectional area of myofibers with centrally located nuclei. The novelty of our FI model lies in the combined methods’ design procedures (cooling method, pressure, and duration) simulating a severe traumatic injury model.

The goal of this study was thus to characterize the inflammatory and reparative responses following cryoinjury through histological inspection, in vivo functional muscle evaluation, and analysis of transcriptional expression of genes involved in inflammatory and pro-resolving stages of skeletal muscle regeneration. In our model, the injured area displayed massive cell death with a loss of ≥15% of muscle mass. In a recent study, Anderson et al. demonstrated that this percentage represents a critical threshold point for failure of the regenerative process characterized by persistent fibrosis and inflammation [3].

Motor coordination and balance task performances revealed that our critical-sized cryoinjury was associated with loss of function and produced significant and durable deficits in animal ability until the end of the observation period.

We also investigated the temporal pattern of leukocyte infiltration, a hallmark of local inflammation [42]. Indeed, histological analysis revealed the presence of infiltrating neutrophils accompanied by upregulation of genes associated with immune cell chemotaxis early after the FI. Increased infiltration of CD68^+^ macrophages was likewise detected. Phagocytic macrophages sense dsDNA released from dying cells via the cytosolic cGAS/STING/IRF3 pathway [43]. In our cryoinjury model, type I IFNs and ISG transcripts were upregulated early after the muscle injury, thus confirming previous studies that have shown cGAS/STING/IRF3 pathway activation, which promotes inflammatory response, after myocardial ischemic injury [44,45]. Accordingly, the detection of these transcripts was consistent with immunostaining of the freeze-injured muscle, whereby the presence of CD68^+^ cells was observed in the early phase of monocyte infiltration. Similarly to ISGs, we observed that pro-inflammatory chemokine/cytokine Il-6, Ccl2/Mcp1, and Tnf-α transcripts were upregulated at day 1 after muscle injury. A remarkable IL-6 expression was observed at day 1 post-FI compared to other injury models [22]. Notably, CCL2 and IL-6 initiate the recruitment of a variety of multipotent mesenchymal progenitor cells as fibro–adipogenic progenitor cells (FAPs) and muscle-resident stem cells [46,47], while TNF-α supports the inflammatory response and the early myogenic events after injury [23]. Anti-inflammatory Il-10, CD206, and Arg1 mRNA markers were also significantly increased, reaching an expression peak at 4 days post-injury. Increased expression of the mannose receptor (CD206) is a phenotypic hallmark of M2-biased macrophages [48], and IL-10 is involved in increased CD163 expression and in M1 to M2 macrophage transition [49]. Furthermore, owing to increased levels of arginase 1, macrophages play a key role in synthesizing collagen and extracellular matrix [50]. These results fit with the presence of macrophages biased towards a pro-reparative phenotype, staining positively for expression of the scavenger receptors CD163 and CD206 in the injured muscles at 4 days post-FI.

Our FI model displayed persistent CD68^+^ macrophage infiltration at late time points post-FI. Sustained macrophage infiltration, clearly demonstrated at long-term post-injury in volumetric muscle loss (VML) defects, was depicted in injuries with unresolved inflammation and suboptimal regenerative outcomes [3,51]. In addition to inflammatory phenotype, the FI-damaged skeletal muscle was characterized by collagen deposition (i.e., replacement fibrosis). The results from this study indicated that FI determined fibrosis, which was clearly visible in the Sirius Red staining imaging until 25 days post-FI. Furthermore, molecular and cellular analyses showed both that the eMHC gene was still transcribed and the persistence of smaller myofibers centrally nucleated until 25 days post-FI compared with uninjured tissues. This prolonged eMHC expression could likely reflect a continued effort at muscle regeneration. The delayed eMHC expression was reported for critically sized VML injuries [3] and bupivacaine-induced injury [52], whereas other injured skeletal muscles do not usually express eMHC 7 days post-damage [53,54,55]. The retaining of Pax7 mRNA expression and positive IHC staining for PAX7 at late time points indicated that the quiescent MuSCs were still activated and poised to differentiate towards newly regenerating myofibers.

Our FI model induces a critical loss of muscle mass that causes the deposition of fibrotic tissue and an inability to restore motor functions completely. These peculiar aspects make it an ideal model for the study of stem-cell-based therapeutic protocols of regeneration in ablative injuries with chronic functional deficits. The stem cells could facilitate the repair of muscle injuries through inflammatory response modulation. In regenerative medicine, exogenous mesenchymal cells have become an alternative source of stem cells that could boost the repair of tissue injuries following transplantation. hAMSCs promote endogenous tissue repair and regeneration through paracrine functions on innate immune system regulation [56].

This work showed that RAW 264.7 M1 macrophages and hAMSCs influence each other via soluble factors. We demonstrated that hAMSCs promote M1 macrophages switching to an anti-inflammatory phenotype, as evidenced by the downregulation of Tnf-α and Nos2 inflammatory markers and upregulation of the anti-inflammatory IL-10 cytokine expression. To our knowledge, our data also showed for the first time that mouse LPS-primed macrophages promoted hAMSCs to express immunomodulatory factors such as COX-2, IDO, and HGF. In contrast, the unstimulated M0 RAW 264.7 macrophages have no effects. Our study, therefore, supported the concept that MSCs are mainly activated by inflammatory cytokines’ milieu, as reported by Lim et al., showing that inflammatory factors such as IFN-β enhanced expression of the immunoregulatory factor IDO at the transcript level [57]. Similarly, we observed an increase in the expression levels of the Ido mRNA when hAMSCs were exposed to pro-inflammatory M1 macrophages. Saldaña and colleagues also reported that primed MSCs co-cultured with macrophages affected TNF-α and IL-10 levels by secretion of the anti-inflammatory prostaglandin E2 (PGE2) [18]. We could assume a similar effect as Cox-2 mRNA levels increased in primed hAMSCs co-cultured with M1-biased macrophages.

By looking at the potential application of hAMSCs, the results demonstrating the crosstalk between hAMSCs and macrophages, particularly the downregulation of TNF-α, contribute to understanding how to set up a more favorable microenvironment for tissue repair mechanisms. The administration of hAMSCs could help to manage the inflammation in a timely way, enhancing the frequency of the M2-biased macrophages within the injured muscle tissue and leading to increased kinetics of muscle healing and repair. It might even be crucial that transitory suppression of IFN-mediated innate immune response could reduce the inflammation, enabling better transplantation of mesenchymal cells.

The optimal timing of transplantation, which depends on the local inflammatory environment of individual diseases and trauma, remains controversial [58,59,60,61,62,63]. Different inflammation statuses, which MSCs encounter after transplantation, might lead to distinct MSC engraftment and therapeutic efficiencies. Overall, our findings suggest that MSCs delivery may be more effective when administered 4 days after FI.

The limitation of the study, however, is that transplantation experiments of stem cells were not performed. The potential therapeutic applications of human mesenchymal stem cells and their clinical significance must be explored in vivo to assess how they can affect the endogenous regenerative process during tissue repair.

The ongoing phase of our research activity deals with transplantation experiments of hAMSCs in our in vivo freeze-injured muscle model. It might also be intriguing to observe whether the inflammatory and reparative phases of muscle healing display different molecular and morphological features due to human mesenchymal stem cell administration.

## Figures and Tables

**Figure 1 biomedicines-12-00030-f001:**
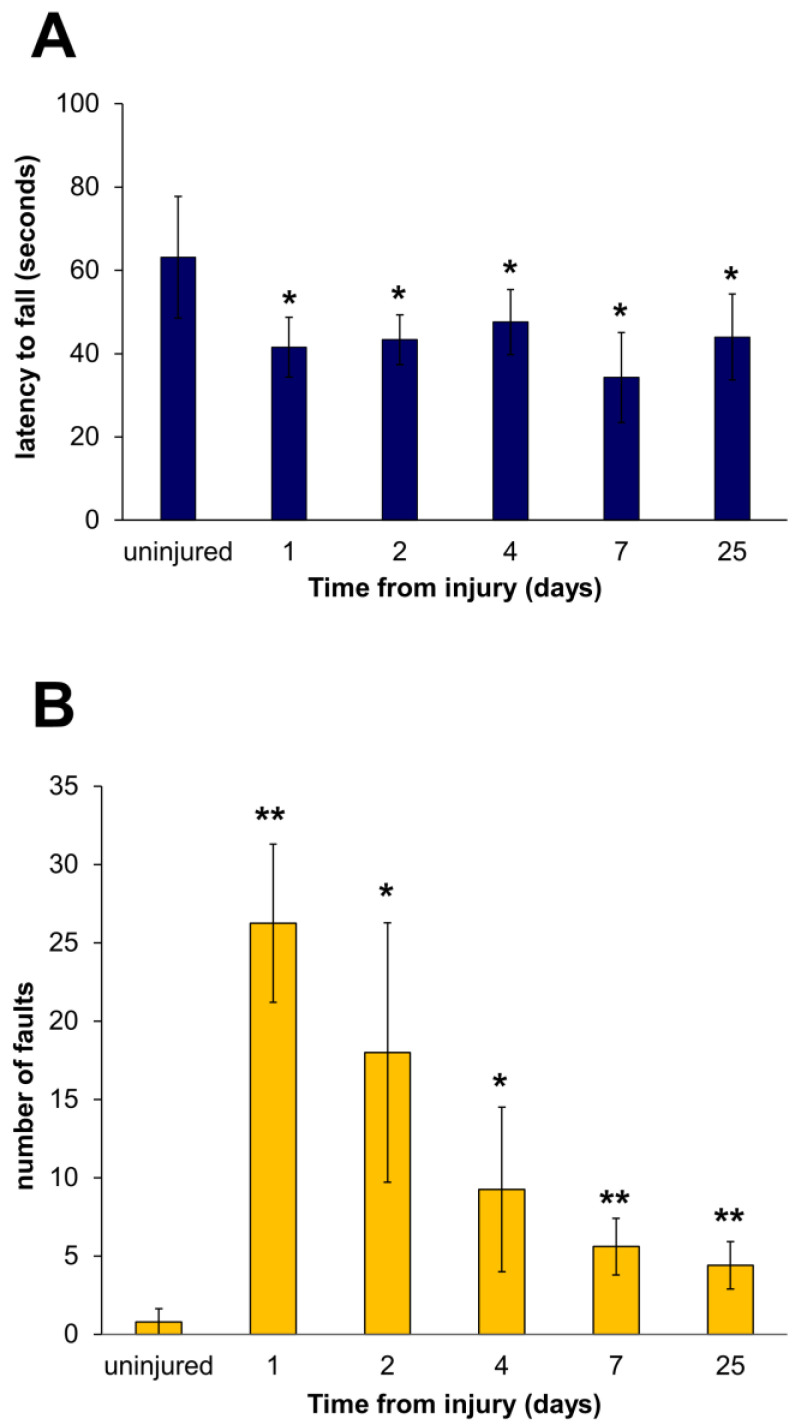
FI protocol produced significant deficits in the animal’s functional ability. (**A**) Latency to fall in rotarod test at 1, 2, 4, 7, and 25 days after injury. The average period (±SD) of each group (4 mice) for the 3 repeated tests is shown. Uninjured mice represent reference control with 100% functionality. (**B**) The average number of paw slips (±SD) for 3 repeated tests is represented by a histogram chart. Uninjured shows the minimum number of slips by the healthy group. One-way ANOVA with a post hoc test. *n* = 4 mice in each group. * *p* < 0.05 and ** *p* < 0.001 vs. uninjured.

**Figure 2 biomedicines-12-00030-f002:**
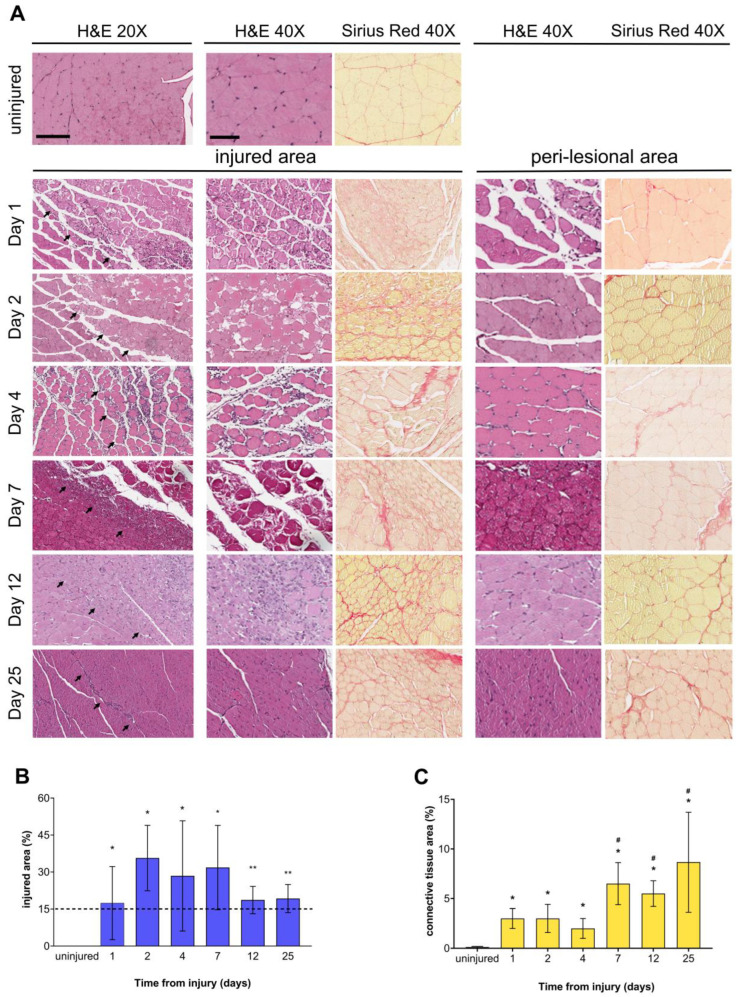
FI provoked histological and morphometric modifications in mouse *biceps femoris* muscle tissue. (**A**) Representative H&E and Sirius Red staining (for collagen deposits) of injured muscle tissue sections at each sampling time point after FI. Both uninjured tissues and areas peripheral to injury were examined. Scale bars: 100 µm (20×), 200 µm (40×). Arrows show the damage boundary line. (**B**) Injury size assessment at each sampling time point after FI (data are expressed as the percentage ratio of damaged area vs. total area); the dotted line indicates the critical threshold point (15%) of the muscle defect as described by Anderson et al. [3]. (**C**) Collagen (connective tissue) quantification in the muscle tissue sections at each time point after FI. Graph bars represent the percentage of the injured area occupied by connective tissue. Data are expressed as mean ± SD. ANOVA with a post hoc test; *n* = 4 mice in each group. * *p* < 0.05 and ** *p* < 0.001 vs. uninjured and # *p* < 0.05 vs. day 4.

**Figure 3 biomedicines-12-00030-f003:**
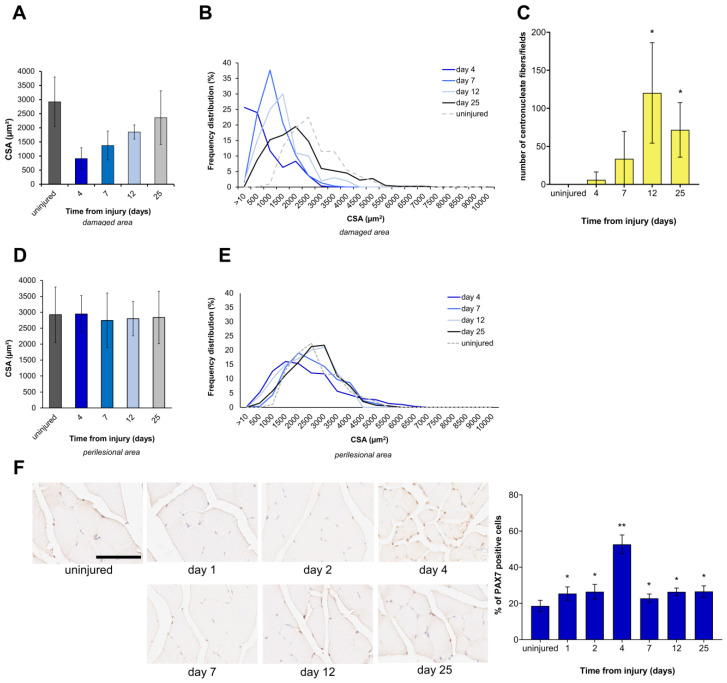
Muscle regeneration parameter assessment after FI. Histograms representing quantification of cross-sectional area (CSA) of myofibers, expressed in µm^2^; frequency distribution of myofiber CSA. (**A**,**B**) Freeze-injured site, (**D**,**E**) region surrounding the lesion, (**C**) quantitative evaluation of muscle fibers in transverse section with centrally positioned nuclei in the regenerating area. (**F**) IHC identification of PAX7^+^ satellite cells (brown reaction) at each time point after FI. In the injured samples, images were acquired at the periphery of the lesioned area. The scale bar represents 100 µm. Relative histograms representing the percentage of PAX7^+^ cells were generated from the same data. n = 4 mice in each group. Data are expressed as mean ± SD. * *p* < 0.05 and ** *p* < 0.001 vs. uninjured.

**Figure 4 biomedicines-12-00030-f004:**
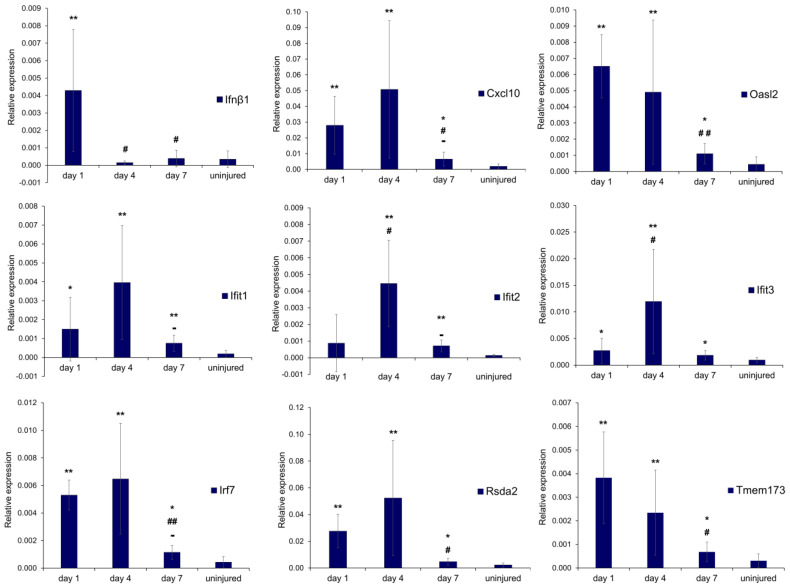
FI activates IRF3-dependent cytosolic DNA sensing and inflammatory response. Expression levels of Ifnβ1, Cxcl10, Ifit1, Ifit2, Ifit3, Oasl2, Irf7, Rsda2, Tmem173. Uninjured muscles were used as controls. One-way ANOVA with a post hoc test. Data are expressed as mean ± SD. *n* = 4 mice in each group. * *p* < 0.05 and ** *p* < 0.001 vs. uninjured and # *p* < 0.05 and ## *p* < 0.001 vs. day 1 and - *p* < 0.05 vs. day 4.

**Figure 5 biomedicines-12-00030-f005:**
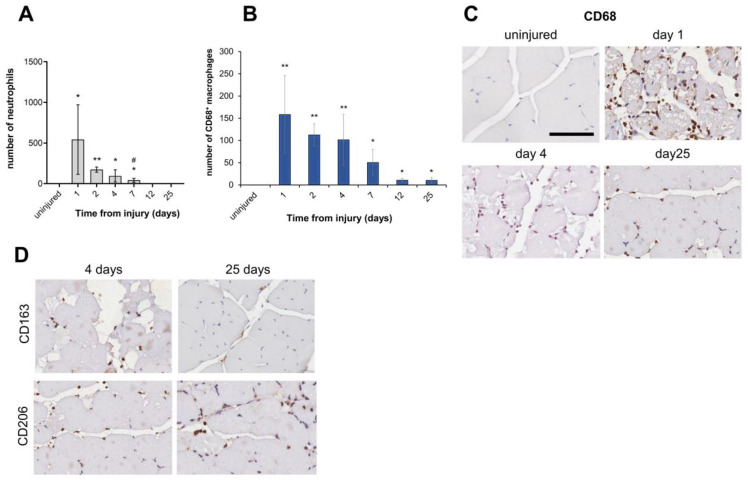
FI induced substantial inflammatory cell infiltration in damaged muscle tissue areas. (**A**) Quantitative evaluation of neutrophils after FI compared with uninjured control. Data are expressed as mean ± SD. (**B**) Quantitative evaluation of CD68^+^ macrophages. Data are expressed as mean ± SD. (**C**) CD68^+^ immunohistochemical staining of macrophage cells (brown reaction). Representative muscle sections in uninjured and injured tissue areas at early (day 1), middle (day 4), and late (day 25) stages after FI; (**D**) CD163^+^ and CD206^+^ immunohistochemical staining of macrophage cells (brown reaction). Representative muscle sections in injured tissue area at 4 and 25 days after FI, respectively (40×, scale bar 100 µm). Data are expressed as mean ± SD. *n* = 4 mice in each group. * *p* < 0.05 and ** *p* < 0.001 vs. uninjured and # *p* < 0.05 vs. day 2.

**Figure 6 biomedicines-12-00030-f006:**
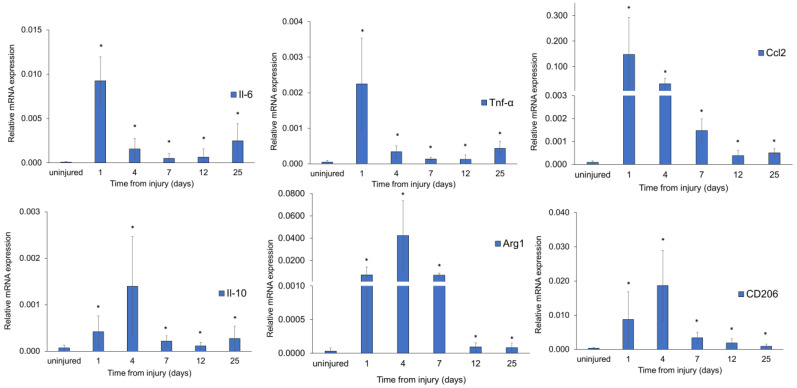
FI-induced expression of pro-inflammatory and anti-inflammatory markers in damaged muscle tissue. Expression levels of pro-inflammatory (Tnf-α, Il-6, Ccl2) and anti-inflammatory (Il-10, Arg1, and CD206) markers. Uninjured mice were used as controls. Statistical significance was evaluated by one-way ANOVA with a post hoc test. *n* = 4 mice in each group. * *p* < 0.05 vs. uninjured.

**Figure 7 biomedicines-12-00030-f007:**
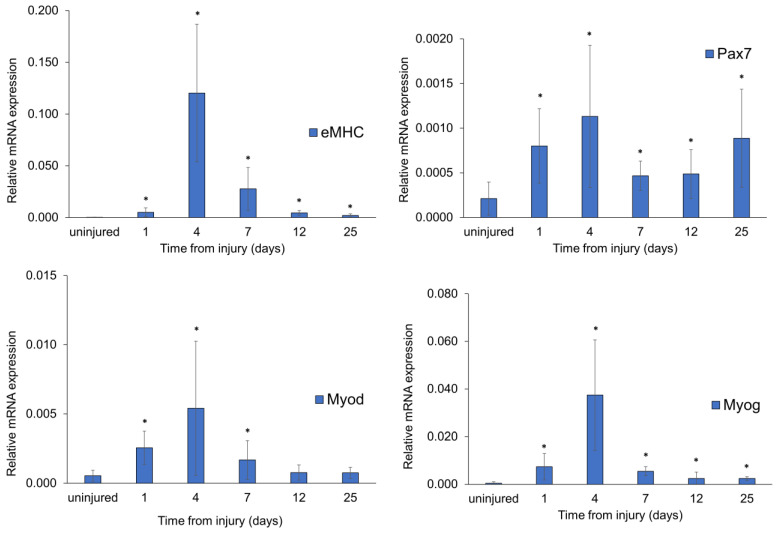
FI-induced expression of muscle regeneration markers in damaged mice. Expression levels of myogenic regeneration markers, eMHC, Pax7, Myod, and Myog. Uninjured mice were used as controls. Statistical significance was evaluated by one-way ANOVA with a post hoc test. Data are expressed as mean ± SD. *n* = 4 mice in each group. * *p* < 0.05 vs. uninjured.

**Figure 8 biomedicines-12-00030-f008:**
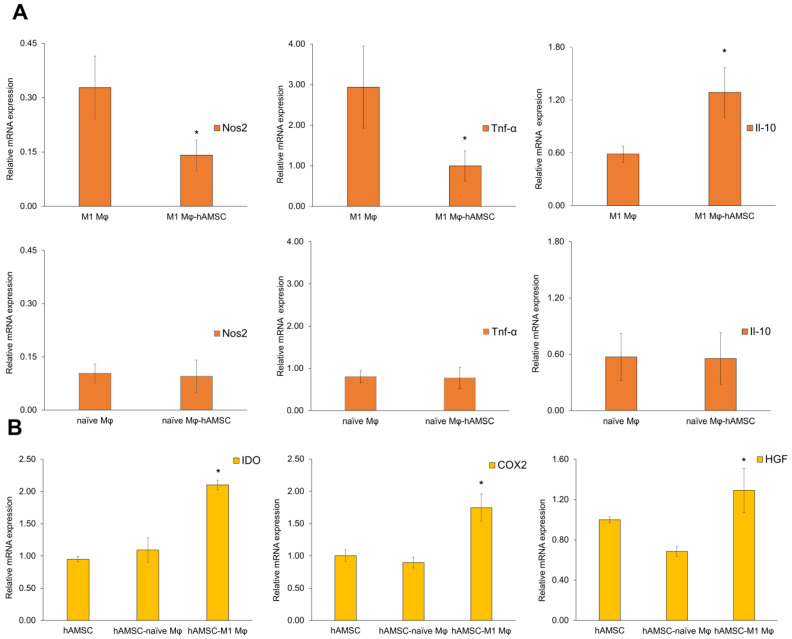
(**A**) hAMSCs affected mRNA expression of pro-inflammatory and anti-inflammatory markers in M1-polarized mouse RAW 264.7 macrophages. mRNA expression levels of pro-inflammatory and anti-inflammatory markers (Nos2, Tnf-α, Il-10) in naïve (naïve Mφ) and M1-polarized macrophages RAW 264.7 (M1 Mφ). Statistical significance was evaluated by Student’s *t*-test. * vs. M1 Mφ. (**B**) M1-polarized (M1 Mφ) RAW 264.7 macrophages were able to trigger immunomodulatory molecule transcription in hAMSCs. mRNA expression levels of immunomodulatory markers (IDO, COX-2, and HGF) in primed hAMSCs compared with hAMSCs alone. Statistical significance was evaluated by one-way ANOVA with a post hoc test. Data are expressed as mean ± SD. *n* = 4 mice in each group. * *p* < 0.05 vs. hAMSCs.

## Data Availability

All data generated or analyzed during this study are included in this published article and Appendix A files.

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
