# Peer review of "Severely Damaged Freeze-Injured Skeletal Muscle Reveals Functional Impairment, Inadequate Repair, and Opportunity for Human Stem Cell Application"

_biomedicines, 2023, doi:10.3390/biomedicines12010030_

Round 1

Reviewer 1 Report

Comments and Suggestions for Authors

In the manuscript entitled [Severely Damaged Freeze-Injured Skeletal Muscle Reveals Functional Impairment, Inadequate Repair, and Open Up for Human Stem Cells Application] by Fioretti et al., the authors studied the impairment of skeletal muscle by freeze injury. Also, they studied cytokine expression using a coculture of macrophages and human amniotic mesenchymal stem cells (hAMSCs). The manuscript is fairly well-written; however, there needs corrections. The comment will be of help for improvement.

Please state why the authors selected hAMSCs. Describe the limitation of the study. Transplantation experiments of stem cells are not performed.

Other comments

Line 36, describe what function the authors mean.

Line 191, the title of 2.8, PAX7+ Satellite cells is better.

Line 232 and 246, please check the concentration of penicillin and streptomycin. Are they correct?

In the manuscript, [not injured] might be better to describe as [uninjured].

Line 544, correct [stimolated].

[Figures]

In many figures, many lower error bars are hidden in the boxes of data except in Figure 8. For example, please see Figure 1A and B. It is advisable that they should be visible like Figure 8.

In Figure 1 legend, # is not in the actual Figure. In Figure 1B, foults may be misspelled.

In Figure 2A, unify the font size. [H&E 20X] is too large.

In Figures 4 and 6, the sizes of cytokine names should be shown as the larger size.

[There are multiple occasions regarding spacing problems.]

Line 214, add space after ;.

Line 231, add space after 2mM.

Line 243, add space after 20.

Line 270, add space after between 10 and sec.

Line 355, add space before indicating.

Comments on the Quality of English Language

Please see the above comments.

Author Response

REPLY BY AUTHORS

Manuscript “Severely Damaged Freeze-Injured Skeletal Muscle Reveals Functional Impairment, Inadequate Repair, and Open Up for Human Stem Cells Application”

Biomedicines-2760875

Fioretti D., Ledda M., Iurescia S. et al.

Reviewer 1

Inizio modulo

Comments and Suggestions for Authors

In the manuscript entitled [Severely Damaged Freeze-Injured Skeletal Muscle Reveals Functional Impairment, Inadequate Repair, and Open Up for Human Stem Cells Application] by Fioretti et al., the authors studied the impairment of skeletal muscle by freeze injury. Also, they studied cytokine expression using a coculture of macrophages and human amniotic mesenchymal stem cells (hAMSCs). The manuscript is fairly well-written; however, there needs corrections. The comment will be of help for improvement.

Reply:  We sincerely thank the reviewer for the positive feedback and for his comments on our manuscript.

All revisions to the manuscript has been highlighted in yellow.

Please state why the authors selected hAMSCs. Describe the limitation of the study. Transplantation experiments of stem cells are not performed.

Reply: We thank the reviewer for comments on scientific background. We have revised the Introduction section incorporating the requested information also providing a reference. The limitation of the study has been described in the Discussion section.  In this section we also underlined that the ongoing phase of our research activity deals with transplantation experiments of hAMSCs in our freeze injury animal model.

Other comments

Line 36, describe what function the authors mean.

The sentence has been ... “function” has been replaced with “its mechanical behavior”

Line 194, the title of 2.8, PAX7+ Satellite cells is better.

The title of 2.8 was amended accordingly.

Line 236 and 251, please check the concentration of penicillin and streptomycin. Are they correct?

The concentration of antibiotics have been amended.

In the manuscript, [not injured] might be better to describe as [uninjured].

[not injured] has been replaced with [uninjured] throughout the entire manuscript.

Line 551, correct [stimolated].

[stimolated] has been replaced with [stimulated]

[Figures]

In many figures, many lower error bars are hidden in the boxes of data except in Figure 8. For example, please see Figure 1A and B. It is advisable that they should be visible like Figure 8.

In Figure 1 legend, # is not in the actual Figure. In Figure 1B, foults may be misspelled.

In Figure 2A, unify the font size. [H&E 20X] is too large.

In Figures 4 and 6, the sizes of cytokine names should be shown as the larger size.

Reply: Thank you for highlighting these errors in Figures. These have been amended accordingly.

[There are multiple occasions regarding spacing problems.]

Line 214, add space after ;.

Line 231, add space after 2mM.

Line 243, add space after 20.

Line 270, add space after between 10 and sec.

Line 355, add space before indicating.

Reply: Thank you for highlighting these errors. We have fixed spacing problems in the manuscript.

Reviewer 2 Report

Comments and Suggestions for Authors

The paper entitled „Severely Damaged Freeze-Injured Skeletal Muscle Reveals  Functional Impairment, Inadequate Repair, and Open Up for Human Stem Cells Application “ by Fioretti et al. is very well written, so I recommend to be published in Biomedicines after a minor revision of the manuscript.

1.

Is there a valid reason you used a metal probe that was cooled with liquid nitrogen, and not, for example, cooled with dry ice, to damage the skeletal muscle? Do you have information about the temperature of the metal probe during exposure (I assume at least -180 °C)? Furthermore, it is not clear to me why you used such a low temperature to cause an injury when such low temperatures are rarely found anywhere in nature, and in industry only in the case of some malfunction when compressed gases leak. Can you explain?

2.

In chapter 3. Results, the authors describe the methods they used (in several places, eg lines: 267-269; 274-276; 303-305; 377-380, etc.). Given that the methods in this paper are well described in Chapter 2., I believe it is unnecessary to describe them again, so I suggest to the authors that, where possible, they should not describe the methods in the Results section.

3.

A few typos can be found:

-          In line 95: “24g” correct to “24 g”

-          In line 105: “10sec” correct to “10 sec”

-          In line 124: “15mm” correct to “15 mm”

-          In line 164: “3μm” correct to “3 μm”

-          In line 234: “CO2” correct to “CO2

Author Response

REPLY BY AUTHORS

Manuscript “Severely Damaged Freeze-Injured Skeletal Muscle Reveals Functional Impairment, Inadequate Repair, and Open Up for Human Stem Cells Application”

Biomedicines-2760875

Fioretti D., Ledda M., Iurescia S. et al.

Reviewer 2Inizio modulo

Comments and Suggestions for Authors

The paper entitled “Severely Damaged Freeze-Injured Skeletal Muscle Reveals  Functional Impairment, Inadequate Repair, and Open Up for Human Stem Cells Application “ by Fioretti et al. is very well written, so I recommend to be published in Biomedicines after a minor revision of the manuscript.

Reply: We sincerely thank the reviewer for the positive comments on our manuscript. We revised the manuscript accordingly to reviewer’s comments.

All revisions to the manuscript has been highlighted in yellow.

1.

Is there a valid reason you used a metal probe that was cooled with liquid nitrogen, and not, for example, cooled with dry ice, to damage the skeletal muscle? Do you have information about the temperature of the metal probe during exposure (I assume at least -180 °C)? Furthermore, it is not clear to me why you used such a when such low temperatures are rarely found anywhere in nature, and in industry only in the case of some malfunction when compressed gases leak. Can you explain?

Reply: We thank the reviewer for the constructive comments. We performed the muscle injury with liquid nitrogen to induce an ‘inclusive’ injury that could impact multiple aspects of the muscle architecture, with the aim to determine a critically sized non-healing defect, that is not achievable with the dry ice cooling. To our knowledge, there have been no reports of muscle freeze injuries describing the temperature of the metallic probes used for the procedure. Our injury model stems from liquid nitrogen cryosurgery of musculoskeletal tumors and, in general, of malignancies, that leads to cell necrosis and destruction of the basal lamina, where satellite cells are located.   

As the temperature of liquid nitrogen is -196°C, it can be assumed that the probe temperatures achieved < -89 degrees Celsius.

The qualitative histological observation of the entire freeze-injured muscle was actually used to evaluate the severity of this traumatic injury. Indeed, we obtained a massive cell death with a loss of 15% of muscle mass until the end of the experiment. In our model the low temperature was used to obtain a nonhealing muscle defect, with deficits in muscle structure and strength and chronic functional impairment, that mimic an injury induced by crushing and other traumatic physical injury.

2.

In chapter 3. Results, the authors describe the methods they used (in several places, eg lines: 267-269; 274-276; 303-305; 377-380, etc.). Given that the methods in this paper are well described in Chapter 2., I believe it is unnecessary to describe them again, so I suggest to the authors that, where possible, they should not describe the methods in the Results section.

Reply: we thank the reviewer for the comments and suggestions. We have addressed the comments as follows:

Paragraph 3.1.  In this paragraph, we stated the description of method we have used to obtain “a traumatic localized muscle injury” as it combined methods design procedures (cooling method, pressure and duration) simulating a severe traumatic injury model and represented the novelty of our FI animal model. For this reason, we believe it is important to describe the method in this paragraph.

We then amended the following paragraphs accordingly to reviewer’s suggestions (lines, 279-281, 307-309, 382-384, 506-507) avoiding to describe again the methods in other Results section.

The sentence at lines 303-305 was moved to lines 153-156 in the section “2. Materials and Methods”.

3.

A few typos can be found:

-          In line 95: “24g” correct to “24 g”

-          In line 105: “10sec” correct to “10 sec”

-          In line 124: “15mm” correct to “15 mm”

-          In line 164: “3μm” correct to “3 μm”

-          In line 234: “CO2” correct to “CO2

Reply: Thank you for highlighting these errors. These have been amended accordingly.